# *N*-(2-Hydroxyphenyl)-2-Propylpentanamide (HO-AAVPA) Induces Apoptosis and Cell Cycle Arrest in Breast Cancer Cells, Decreasing GPER Expression

**DOI:** 10.3390/molecules29153509

**Published:** 2024-07-26

**Authors:** Berenice Prestegui Martel, Alma Delia Chávez-Blanco, Guadalupe Domínguez-Gómez, Alfonso Dueñas González, Patricia Gaona-Aguas, Raúl Flores-Mejía, Selma Alin Somilleda-Ventura, Octavio Rodríguez-Cortes, Rocío Morales-Bárcena, Alberto Martínez Muñoz, Cesar Miguel Mejia Barradas, Jessica Elena Mendieta Wejebe, José Correa Basurto

**Affiliations:** 1Laboratorio de Diseño y Desarrollo de Nuevos Fármacos e Innovación Biotecnológica, Escuela Superior de Medicina, Instituto Politécnico Nacional (IPN), Plan de San Luis y Díaz Mirón, Ciudad de México 11340, México; berequimica@hotmail.com (B.P.M.); albert.wsker@gmail.com (A.M.M.); cmmejiab@hotmail.com (C.M.M.B.); 2Subdirección de Investigación Básica, Instituto Nacional de Cancerología, Ciudad de México 14080, México; celular_alma@hotmail.com (A.D.C.-B.); dguadalupeisabel@yahoo.com.mx (G.D.-G.); alfonso_duenasg@yahoo.com (A.D.G.); mobarobiol@yahoo.com.mx (R.M.-B.); 3Instituto de Investigaciones Biomédicas, Universidad Nacional Autónoma de México/Instituto Nacional de Cancerología, Ciudad de México 04510, México; 4Laboratorio de Inflamación y Obesidad, Escuela Superior de Medicina, Instituto Politécnico Nacional, Plan de San Luis y Díaz Mirón, Ciudad de México 11340, México; patriciagaona44@gmail.com (P.G.-A.); raflores@ipn.mx (R.F.-M.); octaviordzc@yahoo.com.mx (O.R.-C.); 5Centro de Investigación Biomédica, Fundación Hospital Nuestra Señora de la Luz I.A.P., Ezequiel Montes 135, Tabacalera, Ciudad de México 06030, México; som.ven10@outlook.com; 6Centro Interdisciplinario de Ciencias de la Salud-Instituto Politécnico Nacional (CICS-IPN), Ciudad de México 11340, México

**Keywords:** breast cancer, apoptosis, HDAC8, GPER, triple-negative breast cancer, cell line

## Abstract

In this work, we performed anti-proliferative assays for the compound *N*-(2-hydroxyphenyl)-2-propylpentanamide (HO-AAVPA) on breast cancer (BC) cells (MCF-7, SKBR3, and triple-negative BC (TNBC) MDA-MB-231 cells) to explore its pharmacological mechanism regarding the type of cell death associated with G protein-coupled estrogen receptor (GPER) expression. The results show that HO-AAVPA induces cell apoptosis at 5 h or 48 h in either estrogen-dependent (MCF-7) or -independent BC cells (SKBR3 and MDA-MB-231). At 5 h, the apoptosis rate for MCF-7 cells was 68.4% and that for MDA-MB-231 cells was 56.1%; at 48 h, that for SKBR3 was 61.6%, that for MCF-7 cells was 54.9%, and that for MDA-MB-231 (TNBC) was 43.1%. HO-AAVPA increased the S phase in MCF-7 cells and reduced the G2/M phase in MCF-7 and MDA-MB-231 cells. GPER expression decreased more than VPA in the presence of HO-AAVPA. In conclusion, the effects of HO-AAVPA on cell apoptosis could be modulated by epigenetic effects through a decrease in GPER expression.

## 1. Introduction

Breast cancer (BC) is one of the leading causes of death among women aged 35–64 years old worldwide [1]. Multiple factors are associated with the genesis and development of BC, including estrogens and their receptors [2] (such as nuclear estrogen receptor (ERα/β) and transmembrane G protein-coupled estrogen receptor (GPER) [3,4]) as well as epigenetic alterations [5]. Some BC cells lack ERs, and their increased proliferation is modulated by GPER, which explains why triple-negative breast cancer (TNBC) continues to grow despite treatment with ER modulators [6,7,8]. GPER triggers intracellular pathways induced by estrogens [9], such as epidermal growth factor receptor activation, which increases intracellular levels of cyclic adenine monophosphate, induces calcium mobilization, and activates the mitogen-activated protein kinase [10,11] and phosphatidyl inositol 3 kinase pathways [12]. The genomic response to estrogens is associated with ligand-activated transcription factors via ERs, whereas non-genomic signaling involves events for stimulating intracellular pathways that indirectly modulate gene expression via small GPER agonists [13]. GPER overexpression has been reported in breast [14], endometrium [9], ovary [15], and prostate [16] cancers. In the case of BC, the presence of GPER has been associated with poor prognostic factors, such as increased tumor size, a high risk of metastasis, recurrence, and reduced survival rates [17].

Additionally, TNBC overexpresses GPER [17,18,19]; therefore, it has been suggested that GPER antagonists can be an appropriate therapy in this cell line [20,21,22]. GPER is commonly expressed in vitro in BC cell lines, including MCF-7, MDA-MB-231, and SKBR3 [23,24,25]. In MCF-7, GPER and ERs can mediate intracellular pathways related to cancer development and progression [24], and GPER expression depends on the mRNA levels of ERs. Moreover, the MDA-MB-231 and MCF-7 cell lines contain high mRNA levels of GPER [24,25]. SKBR3 and MDA-MB-231 cells are considered ER-negative and have been widely used as in vitro models to test compounds whose therapeutic target is GPER [26,27].

Meanwhile, carcinogenesis and cancer progression are frequently associated with an aberrant state of protein deacetylation by histone deacetylases (HDACs) [28]. HDAC inhibitors (HDACis) induce cancer cell death by increasing the acetylation of histone and non-histone proteins (e.g., α-tubulin, p53, and E2F) [29]. It is known that HDACs are involved in the control and/or regulation of cell survival, proliferation, and differentiation [30]. The aberrant overexpression of HDAC can lead to an epigenetic imbalance associated with cell proliferation in BC [31]. For this reason, HDACis use can inhibit the growth of cancer cells [32]. HDACis have attracted the attention of oncology researchers, and more than 500 clinical trials have recently been initiated. For example, the Food and Drug Administration (FDA) has approved some HDACis, such as suberoylanilide hydroxamic acid (SAHA, vorinostat) and FK228 (romidepsin), among others [33]. SAHA has shown success in the treatment of cutaneous T-cell lymphoma [34]. It is well known that epigenetic modulation decreases GPER expression [35]. GPER antagonism triggers cell apoptosis and G2/M cell cycle arrest in oral squamous cell carcinoma and MCF-7 BC [36,37]. Moreover, there is evidence that the SAHA group induces GPER downregulation [38].

Based on the above, our work group designed the compound *N*-(2-hydroxyphenyl)-2-propipylpentanamide (HO-AAVPA) in silico, derived from valproic acid (VPA), and subjected it to chemical synthesis and biological evaluation [35]. HO-AAVPA had anti-proliferative effects on HeLa (cervix) and A204 (rhabdomyosarcoma) cells and was more potent than VPA [39]. In addition, we observed the anti-proliferative effects of HO-AAVPA on BC cell lines, such as MCF-7, MDA-MB231, and SKBR3 [39].

In this work, we explored the type of cell death caused by HO-AAVPA over short- and long-term treatment, alongside its effect on the cell cycle and GPER expression, which is associated with cell apoptosis and proliferation.

## 2. Results

### 2.1. HO-AAVPA Induced Apoptosis in Breast Cancer Cell Lines

Flow cytometry was used to analyze cell apoptosis in MCF-7 and MDA-MB-231 cells for 5 h, and SKBR3, MCF-7, and MDA-MB-231 cells for 48 h, under three conditions: (1) HO-AAVPA + DMSO treatment, (2) DMSO treatment, or (3) no treatment. The results show that HO-AAVPA in MCF-7 at 5 h induced more apoptosis (68.4%) than no treatment (16.2%) and with DMSO (16.6%) (*p* ˂ 0.05) (Figure 1). VPA showed more apoptotic effects than no treatment (Figure 1). In the same way, MDA-MB-231 cells treated with HO-AAVPA suffered more apoptosis (56.08%) than those without treatment (23.3%) and with DMSO (29.6%) (*p* ˂ 0.05). VPA maintained a similar cell viability to the control groups. Any treatment-induced necrosis cell death of more than 10% is shown in Figure 1. The results at 48 h show that DMSO maintained ≅ 95.3% of living cells, whereas chelerythrine showed apoptotic effects (Figure 2, Figure 3 and Figure 4). The MDA-MB-231 cells treated with HO-AAVPA exhibited an increased rate of apoptosis (*p* ˂ 0.05) compared with those treated with DMSO (Figure 2). HO-AAVPA significantly decreased the percentage of living cells and increased the percentage of apoptotic cells in the MDA-MB-231 cell line. The percentages of cells in early and late apoptosis were 31.4% and 11.7%, respectively (Table 1). VPA and DMSO exhibited similar cell viability in MDA-MB-231 cells (Table 1). HO-AAVPA treatment increased apoptosis in MCF-7 cells, with rates of 37.6% and 17.3% for early and late apoptosis, respectively (Figure 3 and Table 2). VPA and DMSO showed a similar percentage of living cells (Figure 3 and Table 2). Finally, HO-AAVPA decreased the percentage of living cells and increased cell apoptosis in SKBR3 cells with rates of 50.0% and 11.6% for early and late apoptosis, respectively (Figure 4). VPA and DMSO maintain the viability of cells near 87% (Figure 4).

#### G2-M Levels in MDA-MB-231 and MCF-7 Cells Treated with HO-AAVPA

After analyzing the cell apoptosis in the presence of HO-AAVPA, the cell cycle phases of the MCF-7 and MDA-MB-231 cells exposed to HO-AAVPA were evaluated. Cell cycle analysis was performed at 48 h. The MCF7 cells treated with HO-AAVPA showed a higher percentage of S-phase cells (7.1%) than the control group (3.62%) (*p* < 0.0001). The MCF7 cells treated with HO-AAVPA showed a lower percentage of G2/M-phase cells (1.7%) than those without treatment (12.3%) (*p* ˂ 0.0001). MDA-MB-231 cells treated with HO-AAVPA showed lower percentages (2–3%) than those without treatment (11.8%) (*p* < 0.0004). The percentage of sub-G0-phase cells in both cell lines increased by 3.9% and 7.3 compared with the group without treatment, at 0.04% (*p* = 0.0002) for MCF-7 cells and 0.05% (*p* = 0.0004) for MDA-MB-231 cells, respectively. No statistically significant differences were found in the G0–G1 and S stages between the cells treated with HO-AAVPA and the control (Figure 5).

### 2.2. HO-AAVPA Inhibits GPER Expression

GPER expression was measured to determine whether it was associated with the apoptotic effect of HO-AAVPA on BC cell lines. Figure 6 demonstrates that the studied BC cell lines showed different GPER expression patterns under different conditions, whereas actin expression showed the same behavior for 72 h. The results show the GPER presence in either control cells (without treatment) or with DMSO. When the cells were treated with the IC_50_ values of HO-AAVPA or VPA at an equimolar ratio (Table 3), the results indicate that the HO-AAVPA treatment considerably decreased the GPER expression in the BC cell lines (Figure 6). Furthermore, a decrease in GPER expression was observed with VPA. However, there was little GPER, which could be associated with the absence of effects on cell viability and cell apoptosis, unlike HO-AAVPA (Figure 6).

## 3. Discussion

In this research, we explored the role of GPER in the presence of HO-AAVPA, which could explain its apoptotic and anti-proliferative effects on BC cells. This hypothesis was based on our previous study demonstrating that HO-AAVPA inhibits HDAC [40]. In this regard, it is known that HDAC inhibition decreases GPER expression [38]. The results of this study confirm HO-AAVPA’s role as an apoptotic compound in human BC cell lines (Figure 1, Figure 2, Figure 3 and Figure 4) and its capacity to arrest cells in the S phase (Figure 5). In this study, we observed that HO-AAVPA induced a higher level of apoptosis than VPA (68.43 vs. 47.68% in MCF-7 cells and 56.08 vs. 35.54% in MDA-MB-231 cells after 5 h of treatment). This finding aligns with those reported by Mawatari, who treated the SKBR3 BC cell line with 1 mM of VPA and observed an elevation in active caspase-3 levels from 6 to 48 h post-treatment [41]. At 48 h, flow cytometry showed 43.1% of MDA-MB-231 cells in apoptosis (Table 1) and 54.9% for MCF-7 cells treated with HO-AAVPA (Table 2) using its corresponding IC_50_ for 48 h (Table 3). Wawruszak et al.’s study in 2015 [42] reported that treatment with SAHA or VPA in the MCF-7, MDA-MB-231, and T47D cell lines increased the number of apoptotic cells compared with their control. In addition, combining several HDACis and cisplatin increases the number of apoptotic cells, as described in [42]. For example, the percentages of apoptotic cells were 5.81% and 5.89% in MCF-7 cells treated separately using the IC_50_ of VPA and cisplatin, respectively. However, the percentage of apoptotic cells was 21.4% with the combination treatment (VPA + cisplatin) [42]. In this work, approximately twice the amount of apoptotic cells was observed in MDA-MB-231 (43.1%) using only HO-AAVPA, which was perhaps due to its better anti-proliferative effect than VPA [39]. Wawruszak et al. obtained similar results in 2015 when they tested combinations of VPA and cisplatin in other cell lines, including MDA-MB-231 [42]. However, in our study, 54.9% of MCF-7 cells showed apoptosis with HO-AAVPA, which was higher than the percentage reported for VPA [42]. This could be due to HO-AAVPA’s better anti-proliferative effects compared to VPA [39]. In our study, the percentage of apoptotic cells decreased at 48 h compared with 5 h of treatment for MDA-MB-231 and MCF-7 cells. Previous research has demonstrated that VPA induces histone H3 acetylation peaks between 6 and 24 h [43] with the same rise in active caspase-3 levels in the SKBR3 cell line. This work demonstrated HO-AAVPA’s better anti-proliferative capacity since VPA did not induce cell cycle arrest, while HO-AAVPA induced an increase in the S phase in MCF-7 cells and a reduction in the G2/M phase in MCF-7 and MDA-MB-231 cells. These results agree with those reported by Castillo-Juárez et al., which showed that HO-AAVPA caused apoptotic effects in U87-MG (human glioblastoma) and U-2 OS (human osteosarcoma) cells [44]. This experimental evidence supports why HO-AAVPA induces cell apoptosis in cancer cells. However, there is no clear explanation for HDAC inhibition inducing cell apoptosis [40]. VPA induces apoptosis via an intrinsic pathway by increasing the activities of caspase-8 and -9 and Bcl-2, which results in mitochondrial membrane damage and reactive oxygen species (ROS) production [45]. Therefore, the apoptosis induced by HO-AAVPA is also likely intrinsic. Differences in the effect of HO-AAVPA on the cell cycle compared with VPA were observed. While VPA has been demonstrated to induce arrest in the G1 phase, accompanied by an increase in p21 [46], HO-AAVPA induced arrest in the S phase. Reports have indicated that some HDACIs favor damaged DNA due to radiodensity [47]. HO-AAVPA likely induces apoptosis and cell cycle arrest via this pathway. However, the activities of caspase-8 and -9, the Bid/Bax balance, ROS production, and alterations at the checkpoints between the S and G2 phases of the cell cycle must be determined to elucidate HO-AAVPA’s mechanism of action in BC cell lines. Furthermore, this research determined that HO-AAVPA’s effects on GPER expression in the SKBR3, MCF-7, and MDA-MB-231 cell lines were due to possible epigenetic modulations in HDACIs [40]. The results show that GPER expression decreased with HO-AAVPA treatment compared with the control and DMSO-treated cells (Figure 6). The three cell lines under study expressed the GPER protein in a greater proportion in the MDA-MB-231 cells [48]. There is a controversy in the literature about the role of GPER in cell proliferation; however, GPER’s antagonism is associated with cell death [48]. We observed that HO-AAVPA induced the downregulation of GPER and arrest in the S phase of the cell cycle in MCF-7 cells alongside a decrease in the G2/M phase in MCF and MDA-MB-231 cells. Therefore, these results support the proliferative role of GPER in MCF-7 cells. These results suggest that HO-AAVPA’s possible mechanism of action in the three cell lines is by decreasing GPER expression, particularly in the MDA-MB-231 line, which represents an interesting type of cancer, since it is defined as TNBC, and for which treatment is not yet well defined.

## 4. Materials and Methods

### 4.1. Reagents

HO-AAVPA was synthesized by our work group [39]. Briefly, oxalyl chloride (3.71 mL; 34.6 mmol) was added dropwise to VPA (5 g; 34.6 mmol) at 0 °C. The reaction mixture was stirred for 8 h at the same temperature. Then, it was allowed to warm at room temperature under nitrogen to produce 2-propylpentanoyl chloride. The nitrogen flux was continued and then suspended for 15 min each. This alternating process was carried out for 3 h, which was followed by cooling at −5 °C. Hexane (3 mL) and ortho-hydroxy-aniline (3.74 g) were added, and the mixture was stirred for 12 h before adding hexane (10 mL) and sodium bicarbonate (3 g). Subsequently, the mixture was stirred for 3 h. The compound was obtained via distillate hexane extraction, and the solid formed was filtered and washed with hexane (Figure 1).

#### Cell Lines and Culture

The human breast cancer MCF-7, MDA-MB-231, and SKBR3 cell lines were obtained from American Type Cell Culture (ATCC). MCF-S7 and MDA-MB-231 cells were cultured using Dulbecco’s modified Eagle’s medium (DMEM) (Gibco Company, Waltham, MA, USA), a high-glucose medium without phenol red, containing 5% fetal bovine serum (FBS, Gibco Company, USA). SKBR3 cells were cultured in DMEM/F12 medium without phenol red and supplemented with 10% FBS (Gibco Company, USA) and 1× of penicillin/streptomycin (Gibco Company, USA) at 37 °C in a 5% CO_2_ incubator.

### 4.2. Apoptosis via Flow Cytometry

In this study, apoptosis was determined via flow cytometry using the FITC Annexin V Apoptosis Detection Kit (Roche Molecular Biochemicals, Basel, Switzerland) for a short and a long time based on Mawatari et al.’s work, in which they determined that VPA initiates histone acetylation and apoptosis before 6 h of treatment in BC cell lines [41]. Cells in the logarithmic growth phase were selected and rinsed twice in phosphate-buffered saline (PBS). Three groups of cells were seeded in 75 cm^2^ culture flasks at a density of 2.5 × 10^6^ and treated for 5 h and 48 h with (a) the inhibitory concentration 50 (IC_50_) of HO-AAVPA (Table 3) [39], (b) the equimolar concentration of VPA, and (c) 5 μM of chelerythrine, a cell-permeable inhibitor of protein kinase C [49], as a positive control. Additionally, two groups of cells with either dimethyl sulfoxide (DMSO 0.1% as a vehicle) or the medium were incubated as controls without treatment. After, 1 × 10^6^ treated cells were treated with 800 μL of trypsin 0.05%, and 100 μL of annexin buffer 1X, 2 μL of propidium iodide, and 2 μL of annexin V were added. The cells were then incubated in the dark at room temperature for 15 min, and samples were obtained in duplicate. Finally, samples were acquired via flow cytometry (FACS BD Canto II, Beckton Dickinson, Franklin Lakes, NJ, USA) using the FACS Diva V6.0 software. In all cases, 10,000 events were acquired. The results were reported and analyzed in FlowJo V 10.3 (LCC, Ashland, OR, USA) as the percentage of cells in each apoptotic phase (early and late apoptosis, live and necrotic cells).

#### Cell Cycle Assay

To evaluate cell cycle arrest, an initial amount of 1 × 10^5^ cells/well was seeded in a 24-well plate for 48 h under sterile conditions using doxorubicin as a positive control [50]. Four groups were organized under the same conditions used for apoptosis: (a) non-treatment, (b) DMSO (vehicle), (c) HO-AAVPA at IC_50_, and (d) equimolar concentration of VPA. The cells were then harvested into 5 mL flow cytometry tubes, sterile PBS was added, and they were recovered via centrifugation at 1500 rpm for 5 min. The pellets were obtained and then resuspended in 300 μL of 1X PBS at 4 °C. The cells were fixed by adding 700 μL of 90% ethanol drop by drop and with slight agitation (gentle vortex) at 20 °C. Once fixed, the cells were stored at −20 °C overnight. The next day, the cells were washed with 1X PBS at 4 °C to remove the ethanol via centrifugation at 1500 rpm for 5 min. The cell pellets were resuspended in 250 μL of 1X PBS at 4 °C, and 25 μL of pancreatic RNase [2 µg/µL] was added. The mixture was incubated for 30 min at 37 °C. Afterward, 6.5 μL of propidium iodide [1 mg/mL] was added, and the mixture was incubated at 37 °C for 20 min. Finally, the cells were analyzed via flow cytometry (FACS BD Canto II, Beckton Dickinson, USA) using the FACS Diva V6.0 Software. In all cases, 10,000 events were acquired. The results were reported and analyzed in FlowJo V 10.3 (LCC, Ashland, OR, USA) as the percentage of cells in each cell cycle phase (G0/G1, S, G2/M).

### 4.3. Protein Extraction and Western Blot

MCF-7, MDA-MB-231, and SKBR3 cells were cultured for 72 h in 75 cm^2^ culture flasks at 3.0 × 10^6^ per flask and treated with the IC_50_ values of HO-AAVPA (Table 3) reported by Prestegui et al. in 2016 [39], using equimolar concentrations with VPA.

The cells were washed with PBS and centrifuged at 1200 rpm for 5 min, and the proteins were extracted using a radioimmunoprecipitation buffer (150 mM NaCl; 1.0% IGEPAL CA630; 0.5% sodium deoxycholate; 0.1% SDS; 50 mM Tris (pH 8.0)) with proteinase inhibitors (cat. no. P8340; Sigma-Aldrich, St. Louis, MO, USA; Merck KgaA, Darmstadt, Germany). The protein concentration was determined using a bicinchoninic acid assay (BCA-1, Bio-Rad Laboratories, Inc., Hercules, CA, USA), and the integrity was assessed using Coomassie staining. A total of 30 μg of protein was separated via 10% SDS PAGE and transferred onto a polyvinylidene difluoride membrane (cat. no. 1620177; Bio-Rad Laboratories, Inc.). The membrane was blocked with 5% skimmed milk in PBS for 1 h at room temperature and subsequently incubated with antibodies against anti-β actin (1:10,000) (Sigma Aldrich, Merck KGaA) and anti-GPER (1:3000) (PA5-28647) (ThermoFisher, San Francisco, CA, USA) overnight at 4 °C. The secondary antibodies were diluted 1:1000 and incubated for 1 h at room temperature. A chemiluminescent substrate (Clarity Western Enhanced Chemiluminescence Substrate, Bio Rad Laboratories, Hercules, CA, USA) was applied to develop films. Bands were normalized to the loading control β-actin. The bands’ densitometry was measured using ImageJ version 1.50 f (the National Institutes of Health, Bethesda, MD, USA).

### 4.4. Data Analysis

All the experiments were repeated at least three times. The final data were expressed as mean values ± SEM (the standard error of the mean). The means were compared between groups with one-way ANOVA. A *p*-value of <0.05 was statistically significant. The statistical significance between controls and treated samples was analyzed with the GraphPad Prisma 6.0 statistical software (San Diego, CA, USA).

## 5. Conclusions

In conclusion, this study’s results suggest that HO-AAVPA, in TNBC as well as other non-TNBC cells, induces apoptosis and arrest in the S phase of the cell cycle, which could be associated with the inhibition of GPER expression by inhibiting HDACs, as with VPA, but with increased potency.

## Data Availability

The data presented in this study are available in article.

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
