# Peer review of "N-(2-Hydroxyphenyl)-2-Propylpentanamide (HO-AAVPA) Induces Apoptosis and Cell Cycle Arrest in Breast Cancer Cells, Decreasing GPER Expression"

_molecules, 2024, doi:10.3390/molecules29153509_

Round 1

Reviewer 1 Report

Comments and Suggestions for Authors

The manuscript by Berenice et al addresses the anticancer effect of HO-AAVPA breast cancer cells. This manuscript is presenting interesting data however the draft has a number of points that need to be addressed:

  1. It is not clear how authors calculated IC50. I suggest the authors to express data as graphs (not table) indicating statistical significance for all cell lines.
  2. The same is related to table 2 and 3. I suggest the authors to express data as graphs (not table) indicating clearly statistical significance.
  3. Western blotting results are presented as a single experiment on Fig 6. I suggest the authors to express data as graphs from three independent experiments indicating statistical significance.
Comments on the Quality of English Language

.

Author Response

The manuscript by Berenice et al addresses the anticancer effect of HO-AAVPA breast cancer cells. This manuscript is presenting interesting data however the draft has a number of points that need to be addressed:

  1. It is not clear how authors calculated IC50. I suggest the authors to express data as graphs (not table) indicating statistical significance for all cell lines.

Response: Thank you for your suggestion, the data is listed in Table 1 because these results were published by Prestegui-Martel B, et al J Enzyme Inhib Med Chem. 2016;31(sup3):140-149. doi: 10.1080/14756366.2016.1210138, reference 39.

  1. The same is related to table 2 and 3. I suggest the authors to express data as graphs (not table) indicating clearly statistical significance.

Response: IC50 from HO-AAVPA is reported Prestegui-Martel B, et al J Enzyme Inhib Med Chem. 2016;31(sup3):140-149. doi: 10.1080/14756366.2016.1210138, reference 39. Regarding VPA, it was used at a concentration similar to HO-AAVPA as is mentioned in the text. DMSO is the % which dilutes HO-AAVPA and VPA and chelerythrine 50 uM belongs to other reports elsewhere. 

  1. Western blotting results are presented as a single experiment on Fig 6. I suggest the authors to express data as graphs from three independent experiments indicating statistical significance.

Response: Thank you, we have update the Figure 6.

Reviewer 2 Report

Comments and Suggestions for Authors

The authors introduced the mechanism of cell death of N-(2-hydroxyphenyl)-2-propylpentanamide in cancer cells. This study seems interesting, however there are some points needs to be addressed. 

- What is the chemical structure of N-(2-hydroxyphenyl)-2-propylpentanamide? What is the rationale of this study? What about the application of molecular docking to highlight the virtual mechanism of binding? 

- Authors should add the IC50 values of one standard drug for comaprison

- What about testing the enzyme targeting using EGFR ELISA Kit

Comments on the Quality of English Language

Moderate editing of English language required

Author Response

The authors introduced the mechanism of cell death of N-(2-hydroxyphenyl)-2-propylpentanamide in cancer cells. This study seems interesting, however there are some points needs to be addressed. 

- What is the chemical structure of N-(2-hydroxyphenyl)-2-propylpentanamide? 

Response: It has been reported in reference 39.

What is the rationale of this study? 

Response: It is because the N-(2-hydroxyphenyl)-2-propylpentanamide has antiproliferative effects (reference 39), however, despite of additional experimental studies that have been developed (reference 43), it is not clear its pharmacological effects. 

What about the application of molecular docking to highlight the virtual mechanism of binding? 

Response: It has been reported previously in reference 39.

- Authors should add the IC50 values of one standard drug for comparison

Response: VPA was included as a standard drug due to its known as HDACi inhibitor and N-(2-hydroxyphenyl)-2-propylpentanamide is derived from VPA.

- What about testing the enzyme targeting using EGFR ELISA Kit

Response: The main target to explore was GPER due to its relationship with HDAC and GPER expression.

Round 2

Reviewer 1 Report

Comments and Suggestions for Authors

The manuscript by Berenice et al addresses the anticancer effect of HO-AAVPA breast cancer cells. This manuscript is presenting interesting data however the draft has minor points that need to be addressed:

  1. Authors provided additional info how is calculated IC50.
  2. Western blotting results are now presented from three independent experiments indicating significant data on Fig 6. I suggest the authors to move the representative blots on the bottom and align with representative graphs.

Author Response

Referee 1

The manuscript by Berenice et al addresses the anticancer effect of HO-AAVPA breast cancer cells. This manuscript is presenting interesting data however the draft has minor points that need to be addressed:

  1. Authors provided additional info how is calculated IC50.

Response: We have included a phrase at Table 1 “IC50 values were obtained experimentally and published by our work group [39].”

  1. Western blotting results are now presented from three independent experiments indicating significant data on Fig 6. I suggest the authors to move the representative blots on the bottom and align with representative graphs.

Response: Thank you for your suggestion, the Figure 6 has been updated as your suggestio